# Simulation-Based Cybersecurity Testing and Evaluation Method for Connected Car V2X Application Using Virtual Machine

**DOI:** 10.3390/s23031421

**Published:** 2023-01-27

**Authors:** Dae-Hwi Lee, Chan-Min Kim, Hyun-Seok Song, Yong-Hee Lee, Won-Sun Chung

**Affiliations:** 1KATECH, Convergence Security Lab., Pungse-ro 303, Cheonan 330912, Republic of Korea; 2KATECH, Reliability-Certification Research Lab., Pungse-ro 303, Cheonan 330912, Republic of Korea; 3Sambo Motors, Research & Development Team 1, Digital-ro 9, Seoul 08510, Republic of Korea

**Keywords:** connected car security, V2X security, security evaluation scenario

## Abstract

In a connected car, the vehicle’s internal network is connected to the outside through communication technology. However, this can cause new security vulnerabilities. In particular, V2X communication, to provide the safety of connected cars, can directly threaten the lives of passengers if a security attack occurs. For V2X communication security, standards such as IEEE 1609.2 define the technical functions that digital signature and encryption to provide security of V2X messages. However, it is difficult to verify the security technology by applying it to the environment with real roads because it can be made up of other safety accidents. In addition, vehicle simulation R&D is steadily being carried out, but there is no simulation that evaluates security for the V2X application level. Therefore, in this paper, a virtual machine was used to implement a V2X communication simulation environment that satisfies the requirements for the security evaluation of connected cars. Then, we proposed scenarios for cybersecurity testing and evaluation, implemented and verified through CANoe Option.Car2X. Through this, it is possible to perform sufficient preliminary verification to minimize the variables before verifying security technology in a real road environment.

## 1. Introduction

With the emergence of connected cars that provide users with safety and convenience, vehicles that had been closed began connecting to the outside. Vehicles contain a variety of interfaces and, simultaneously, the attack surfaces at which attackers can find access have increased, as shown in Figure 1. Inevitably, there is the growing importance of cybersecurity for connected cars. This is because there may be a direct threat to social safety, including the lives of passengers, if a security threat occurs in vehicle to everything (V2X) communication, a service for the safety of connected cars.

WP.29, the World Forum for Harmonization of Vehicle Regulations under the United Nations Economic Commission for Europe (UNECE), has adopted two regulations related to vehicle cybersecurity, UNR155 (UN Regulation No. 155) and UNR156 (UN Regulation No. 156), in June 2020 [1,2]. UNR155 is a regulation for the cybersecurity management system (CSMS) of vehicles, and it is concerned with constructing CSMS for the responses of manufacturers and partner companies to vehicle cybersecurity threats. ISO/SAE 21434 is the reference standard for UNR155 [3]. It mentions prohibiting companies from selling vehicles or parts in Europe if cybersecurity measures are not met, thus making companies act against growing cybersecurity threats to connected cars.

According to Trend Micro, the critical threat level to the cybersecurity defined in UNR155 is 22% for existing vehicles. It is expected to rise to 43% with the spread of connected cars in the future, as shown in Figure 2. It is forecast that low-level threats among the attack surfaces will disappear altogether, with only threats of a medium level and above remaining [4].

As such, the importance of connected-car cybersecurity is growing worldwide, and the development and verification of technologies for connected-car security requires using real vehicle-based test facilities or real-road driving tests. However, it is difficult for security solution developers to build and use real vehicle-based test facilities. It is difficult to conduct tests and verifications on real roads because they affect other vehicles. V2X simulations can be performed using commercial programs, such as Vector CANoe. However, since vehicles implemented by simulations or road infrastructures, such as Road Side Unit (RSU), are simple virtual nodes implemented by software, they cannot be accessed from the outside, thus limiting their use in developing security technologies.

Therefore, this study has implemented V2X communication simulations of actual road driving environments in South Korea and built a virtual machine environment for interoperation, with external interfaces to overcome the limitations of a conventional simulation equipment. Virtual environment tests can be performed for each V2X driving scenario in the implemented environment. An interface allows communication with the outside by linking the virtual machine with simulation nodes. That is, a test environment close to the real-world environment is built. This allows sufficient verification in a simulation environment before applying the connected-car security technology in the real-world environment.

The contributions of this paper can be summarized as follows:We analyze existing simulation and testing methods from the perspective of security evaluation.We propose requirements for V2X security evaluation.We propose misbehavior-based application level V2X security evaluation scenarios.We implement V2X security evaluation scenarios using CANoe Option.Car2X simulation.

This paper is organized as follows. Section 2 introduces the background on V2X standards and simulations and security testing related work. Section 3 presents the problems of the existing V2X simulation, and also presents the requirements of the implementation for security evaluation. Section 4 contains our V2X security evaluation simulation method using a virtual machine. Section 5 defines the security evaluation scenarios and presents implementation results. Section 6 contains the conclusion.

## 2. Background and Related Work

This section describes the background and related work of the simulation-based V2X security evaluation methods. First, we will describe the background of the analysis and security evaluation methods for the SAE J2735 standard, which defines the message sets used for V2X communication. We also analyze the major V2X simulations suggested by academia and industry and present problems in performing security evaluations in conventional simulations.

### 2.1. Background

#### 2.1.1. V2X Standards

Internet of Things (IoT) is used in various ways to solve the problems of smart cities, and Cooperative Intelligent Transport Systems (C-ITS) can be applied as smart city IoT applications. IoT can provide C-ITS services that operate through inter-working with the cloud [5,6]. Data generated in the C-ITS environment can be used in various ways, and the collected data can be used to estimate travel times within the city [7]. V2X means vehicle to everything and is similar to IoT in that it can communicate with vehicles and other objects. However, V2X is a communication technology that focuses more on safety. Recently, various studies related to road safety and traffic efficiency using V2X based on 5G New Radio (NR) with high data rate, low latency, and a wide coverage are being conducted [8].

Figure 3 is a comparison of ITS stacks according to the V2X standard protocols. The V2X communication protocols used in the U.S. can be broadly divided into IEEE 802.11p-based Wireless Access in Vehicular Environment (WAVE) and Cellular V2X (C-V2X), which is based on cellulars, such as Long Term Evolution (LTE) and 5G. WAVE is a technology for short-range, high-speed communication defined by the IEEE 1609 standard family [9,10,11,12] and has a shorter communication range than the cellular-based C-V2X. C-V2X uses specifications defined by Third Generation Partnership Project (3GPP) in the lower-level physical layer, and Rel 14 [13,14] is used for the LTE-bases V2X, while Rel 16 [15] is used for the 5G-based V2X. On the other hand, WAVE and C-V2X use the WAVE Short Message Protocol (WSMP) defined by IEEE 1609.3 [11] in the higher-level network/transport layer and use V2X message sets defined by the SAE J2735 standard in the application layer. In Europe, ETSI ITS-G5 [16] corresponds to WAVE, and message sets, such as Cooperative Awareness Message (CAM) [17] and Decentralized Environmental Notification Message (DENM) [18], defined by ETSI EN 302 637, are used. Of course, the higher-level ETSI protocol can also be applied to C-V2X. This study provides descriptions focusing on V2X defined in the U.S., and this is the same for the proposed simulation-based V2X security evaluation method.

SAE J2735 is a standard that defines the data frame and date element of message sets used in V2X communication [19]. Messages used for vehicle-to-vehicle (V2V) communication for basic driving status, for vehicle speed, global positioning system (GPS) information, heading, angle, brake, and route information are defined as Basic Safety Messages (BSM). Signal Phase and Timing (SPaT) for transmitting traffic signal information from the signal controller, MAP for transmitting road information such as intersections, RSA for roadside alerts, and Probe Vehicle Data (PVD) for collecting a vehicle’s information via V2I are defined in the SAE J2735 standard. In the recently revised version, additional message sets have been defined for various autonomous driving-related services, such as Toll Advertisement Message (TAM) for tolling and Sensor Data Sharing Message (SDSM) for sensor data sharing. Table 1 describes the main message set of SAE J2735 standard.

BSM is the most basic message in V2X. It broadcasts the surroundings through V2V to inform about the status of surrounding vehicles. Through this, the surrounding vehicles collect information about the pertinent vehicle and can use it in the Advanced Driver Assistance System (ADAS) or autonomous cooperative driving. BSM is divided into Part I data (information that must be essentially included) and Part II data (optional extension information). Part I data include essential information to inform about a vehicle’s current status, such as the speed, GPS position, steering, and transmission information, to its surroundings. Part II data include supplementary information, such as a vehicle’s route information, event information occurring in the vehicle, and the vehicle’s size or type. Figure 4 shows the structure of Part I and Part II of BSM defined in the SAE J2735 standard.

BSM has a very important role in road safety, but location errors often occur because they are GPS-based. To solve this problem, a method of collecting and correcting BSM by RSUs to obtain accurate location information of nearby vehicles has also been proposed [21]. A method to increase the positioning accuracy is also proposed [22].

#### 2.1.2. V2X Security Evaluation Method

In V2X communication, the latency requirement is less than 100 ms because communication is performed between fast-moving vehicles [23,24]. As a major accident can occur if the incorrect data are transmitted, data have to be sent and received quickly. Efficient security is provided using certificate-based digital signatures for V2X communication security. In V2X communication, encryption is used when the vehicle updates the certificate key. Encryption is not provided when sending general SAE J2735-based message sets; instead, the authentication, integrity, and non-repudiation properties are provided through the digital signatures. Since a BSM is a message broadcasted to the surroundings, it is transmitted by applying a digital signature. When a V2X certificate is sent, it includes a signature based on the Security Credential Management System (SCMS) [25] or the IEEE 1609.2.1 [26] standard. Figure 5 shows two cases of hacking that can occur during V2X communication in a C-ITS environment. Attackers can send compromised BSM messages to other vehicles via V2X while driving, or attack the C-ITS infrastructure, such as RSU, to occur events to near vehicles. In this situation, a digital signature can be applied effectively.

The basic V2X security is provided through digital signatures, as described above, but V2X security standards are still under revision, and related research is underway to apply them to actual infrastructures of Cooperative Intelligent Transport Systems (C-ITS). In particular, the application and verification of security technology development in the C-ITS environment require sufficient simulations in a virtual environment before they can be performed on real roads. This is because if a security threat is created for technology verification in the real-world environment, it can lead to an accident.

J. Wang et al. introduce various testing methods to be performed in the development of V2X technology, as well as security tests [27]. J. Wang et al. introduce methods of conformance testing, function testing, performance testing, vehicle gateway testing, penetration testing, accelerated testing, and field testing for a device under test (DuT), such as a vehicle or onboard unit (OBU). They also explain the HIL (hardware in the loop)-based parallel testing method for reducing the cost and risk of field testing. Figure 6 describes the parallel testing method proposed by J. Wang et al. The parallel testing method performs simulations and verification by mapping a vehicle object of the real-world environment to a vehicle object of the virtual environment and implementing multiple vehicles running in the virtual environment. Furthermore, J. Wang et al. define security threat attributes for V2X communication by classifying them into authentication, availability, data integrity, confidentiality, non-repudiation, and real-time constraints. For security evaluation in a virtual environment, it is necessary to build the HIL environment for parallel testing, create threats to the security attributes, and perform tests at the application level to process messages to determine whether the DuT is robust against security threats. It means the ability to process misbehaviors in driving situations. These simulations must be performed to implement and verify the function that can filter misbehavior messages when the vehicle receives them.

### 2.2. Related Work

#### 2.2.1. V2X Simulation Research

Z. Lokaj et al. introduce the C-ITS SIM developed by the Czech Technical University (CTU) in Prague [28]. C-ITS infrastructures are under expansion in Europe as a part of the C-Roads project in Europe, and Z. Lokaj et al. propose simulations for testing the interoperability between the C-ITS infrastructure and vehicles and testing the validity of messages. A C-ITS unit that can communicate with the C-ITS infrastructure from the driving vehicle is installed, and transmitted/received data can be checked using the user interface on a laptop. Tests are provided through the receiving mode, in which messages are received from the surrounding C-ITS infrastructure, and the broadcasting mode sends messages to the surrounding C-ITS infrastructure. In all communications, the ITS-G5 standard [16], which is Europe’s V2X standard, is used, with ETSI TS 103 097 [29] and ETSI 102 941 [30], which are standards for security. However, since this simulation tests only the interoperability and the validity of messages, it is close to a conformance test, and it is not appropriate to perform tests for security threats while driving at the application level.

Veins is an open-source vehicle network simulator [31]. Veins performs simulations based on the event-based network simulator OMNeT++ and the SUMO road traffic simulator. In particular, Veins discloses a misbehavior dataset, VeReMi (Vehicular Reference Misbehavior Dataset), using Veins simulations [32]. The authors of [32] state that misbehavior detection aims to discard malicious messages for other vehicles by analyzing application data. However, the authors of [32] present the limitations of VeReMi: the implemented misbehavior type cannot represent all possible attack types in V2X communication. Therefore, it is impossible to analyze the impact of various attack types in a single simulation data of VeReMi. Furthermore, since attack detection is performed only to detect V2X communications that are not interactive, it is difficult to apply a solution for this problem and perform a security evaluation. This can be solved by implementing a message handler for the V2X messages received. If there is data with which misbehavior detection has been performed, a handler that can process them is implemented, and a security evaluation of the handler can be performed.

#### 2.2.2. Connected Car Security Testing Tools

Spirent developed a V2X emulator to test the V2X functions and performance and test the conformance of the WAVE and C-V2X protocol stacks [33]. It can perform U.S. OmniAir-compliant tests, including messages’ conformity and security function tests [27]. Furthermore, if call simulator equipment is used, WAVE and C-V2X full-stack signals can be generated [34]. By applying a driving scenario, we can build a virtual C-ITS HIL environment through node configuration, in which messages are sent/received. If a HIL environment is built by connecting to a specific DuT, a security evaluation can be performed for the message processing part mounted on the DuT. However, one cannot perform an application-level test or verification of the nodes implemented virtually before product development. Vector CANoe Option.Car2X can implement V2X nodes and directly implement the handler of transmitted/received messages through CAPL [35]. Therefore, simulations for implementing a DuT virtually and performing security evaluations of the message handling part for the DuT can be performed based on security evaluation scenarios using CANoe. In this study, we propose applying security evaluation scenarios of vehicle nodes using CANoe Option.Car2X, and explain this in Section 5. Spirent V2X test simulation provides ITS full stack and can perform simulations similar to reality. However, in order to perform the application level tests, tools capable of the application message handling, such as CANoe, should be used. Since CANoe does not basically provide an application level security test, a security evaluation must be performed through the CANoe-based simulation by linking objects that can generate the application event messages.

## 3. Requirements Analysis

This section propose the problems of existing simulation methods and a list of requirements for the trends analyzed in Section 2.

### 3.1. Problems of Existing Simulation

In the existing simulation studies and tools introduced in Section 2.2, HIL environments can be built. Still, simulations can be performed only for the message conformity testing or security function testing of messages in driving according to the pre-composed scenarios. When performing the application-level V2X security test introduced in Section 2.1.2, we need to consider scenarios where threats occur in driving situations. However, it is difficult to create threats, such as the misbehavior of vehicle nodes, in conventional simulations. The reason is that vehicle nodes in the simulation are virtual nodes implemented by software and move according to the set scenario. Even if a security threat is created, there is no threat-based handler. Therefore, there is a limitation in developing the security technology through the monitoring and analysis of threats. Therefore, the simulation requirements presented in Section 3.2 must be applied to perform a security evaluation for application-level misbehavior. According to ISO/SAE 21434, the reference standard for the UNCEC WP.29 regulations, developing a product equipped with security functions requires the application of continuous solutions for misbehavior detection and handling [3]. Therefore, it is essential to perform security evaluations by applying the requirements for evaluating misbehavior.

### 3.2. Requirements of Implementation

The requirements for the simulation-based security evaluation method for connected cars using virtual machines proposed in this study are as follows.

#### 3.2.1. Acceptance of External Message

It should be possible to receive V2X signals or data and use them in simulations, rather than generating messages only according to the specified scenarios in the V2X communication simulations of connected cars. This applies to most simulation tools that can configure HIL. If data are generated by simply writing and executing scenarios, simulations will be just for acquiring driving data. The CANoe can receive V2X RF signals through the WAVE interface and apply them in simulations. However, to generate misbehaviors using V2X RF signals, we need additional tools and an environment that enable the generation of modified messages for abnormal driving in the transmitting device. In this study, a virtual machine is connected to the simulation nodes of CANoe Option.Car2X, and messages can be sent to the simulation nodes by generating the messages through the virtual machine’s software.

#### 3.2.2. Application of Message Handler

It should be possible to apply a handler to perform different operations, respectively, when a V2X node receives a benign or attack message, such as a misbehavior or threat message, in the simulation environment. This is to test the application-level security. Conventional simulations have implemented only the handler for conformity testing of V2X messages. Still, in this study, the proposed part that processes messages when received at a node is implemented to process misbehavior. Since the user can implement a handler for the messages transmitted/received at the nodes in the CANoe simulation environment, as described above, this paper uses it to implement and use the message handler of nodes.

#### 3.2.3. Creation and Application of Misbehavior Scenarios

The part where the virtual machine creates misbehavior scenario events in the simulations for the security tests of a connected car can be implemented in the message handler. Through this, we can create not only the security evaluation scenarios proposed in this study but also the scenarios to test various misbehaviors that can occur in the future. Furthermore, the created scenario can be sent to a node where the message handler is implemented in the simulation to check whether the security evaluation is performed correctly.

### 3.3. Requirements Analysis

Table 2 compares the existing V2X simulations and security testing tools introduced in Section 2.2. In order to detect and process misbehavior, there should be an application message handler capable of receiving and processing messages using external equipment and tools. In addition, it should be evaluated whether the application message handler of the evaluation target is tolerant to misbehavior in a driving situation. CANoe satisfies most of the implementation requirements presented in Section 3.2. In particular, since the tester can develop the send/receive processor for the message, it is possible to create an abnormal message and implement a misbehavior scenario. However, since CANoe only provides a development environment, message handlers should be implemented to generate misbehavior. Therefore, in this paper, abnormal messages and misbehaviors are implemented based on CANoe. Furthermore, by developing a security evaluation scenario that can evaluate this, it is possible to perform an application-level security evaluation test that can occur during driving.

## 4. Proposed Test Simulation

V2X communication simulation equipment is used when performing tests according to the user’s objective by composing various scenarios in advance because of actual tests’ cost, time, and safety difficulties. However, vehicle nodes or ITS infrastructure nodes implemented in simulations are virtual nodes implemented by software, thus having no characteristics of the hardware. As analyzed in Section 3.3, conventional simulations have their objectives and functions specific to the simulations for the objectives. However, as analyzed in Section 3.3, no simulations reflect the security evaluation requirements. Each node in the simulation is connected to its network inside the simulation only, making it challenging to test weaknesses. Moreover, since they are not connected to the external network, validation tests cannot be performed on security threats that may occur in the real-world environment. Furthermore, because of the difference in the programming language used in the security system to be simulated and tested, it is difficult to apply the security system to the simulation nodes. In this paper, therefore, we describe a method of configuring a simulation environment using a virtual machine to improve the problems of conventional simulations and satisfy the requirements for V2X communication security evaluation.

### 4.1. Configuration of Environment Linked to Virtual Machine

Since each node implemented in a typical commercial V2X communication simulation exists only within the simulation, it is not connected to the external ITS infrastructure network, unlike the ITS components in the real-world environment [34,35]. Furthermore, it can implement only virtual software nodes that do not include operating systems (OS) and security systems and do not have the characteristics of the hardware. In this section, we will configure a simulation environment closer to the real-world environment by linking the virtual machine to the software nodes implemented within the simulation to assign the OS and hardware characteristics to the nodes and connect them to the external network. Through this, the performance of the security functions, such as an intrusion detection system (IDS), can be tested regardless of the programming language of the simulation equipment by sending and receiving data.

As shown in Figure 7, by linking the simulation nodes and the virtual machine, we can create an environment to test attack scenarios in which attacks started at vulnerabilities in the OS, network, and protocols of actual ITS components reaching and affecting the V2X network. Since the simulation test environment proposed in this paper uses a virtual machine, the OS used by each part and system can be freely configured. Moreover, the virtual machine can be linked to multiple simulation nodes according to the tested environment configuration and scenario to configure them as 1:n or n:n, instead of 1:1.

The virtual machine-linked security testing method proposed in this study must use a simulation to implement a handler of transmitted/received messages of the nodes and virtual machine. The simulation used for implementation in this paper used Vector CANoe Pro Option.Car2X, which can directly implement the handler of the transmitted/received messages of the simulation nodes. Figure 8 shows the vehicle and RSU nodes implemented by CANoe Pro Option.Car2X. The RSU nodes are designated virtual machine-linked nodes and are connected to the virtual machine through socket communication. For example, suppose a scenario is applied to perform a security attack on a vehicle node through the ITS infrastructure in the C-ITS environment. In that case, the RSU node becomes a hacked node. Through the hacked RSU, simulations can be performed to verify whether hacking affects the vehicle. A security test can be performed with an actual product by connecting the node with a DuT instead of a virtual machine.

### 4.2. Virtual Machine-Linked Node Test

In this section, to verify the proposed simulation test environment, we tested whether it is possible to send/receive data between the virtual machine and simulation nodes, create vehicle nodes, perform driving, and create events without using the scenario editor in the simulation. The implementation target regions for this test were targeted at specific regions in Incheon and Seoul, which are V2X test beds in Korea. Figure 9 and Figure 10 show the Incheon area and Figure 11 shows the Seoul area.

In conventional simulations, the user must specify vehicle nodes’ routes, speeds, and events using the scenario editor tool, as shown in Figure 9. When the created scenario is executed in the simulation, each node sends and receives V2X messages while driving according to the scenario created with the editor tool. Simulations using the scenario editor have a significant disadvantage in using them for security evaluation: the simulations can only be performed according to the scenarios defined in the editor tool. This makes it difficult to simulate situations for changes due to the external input of the V2X message values, such as GPS and speed, or the appearance of a vehicle that did not exist. In the simulation environment proposed in this paper, it is easy to create unexpected situations that are difficult to create with the scenario editor. Since vehicles’ sudden appearance and events can be implemented in the simulation, various situations required for security testing can be created.

The simulation node receives data from the virtual machine, generates a V2X message according to the standard message specifications, and sends the data to the simulation network. As a result of sending a V2X message from the virtual machine to a simulation node, we found that the vehicle node can be driven without using the simulation editor, as shown in Figure 10. Furthermore, a vehicle event could be generated at a specific location using the BSM Part II data, which contains 13 types of vehicle event information.

The simulation node linked to the virtual machine is implemented to collect data from a specific node so that the BSM generated in the network can be collected. Then, a simple GPS spoofing attack scenario was created. Using this scenario, we collected data for the normal driving situation and data for the GPS spoofing attack situation. Through this, we checked whether the GPS spoofing attack message, fed through the virtual machine, affected the simulation network as intended. As an attack scenario, we implemented a simple GPS spoofing scenario in which the GPS values in the BSM are sent with latitude and longitude values that are not related to the driving route using the same address as the normal driving target vehicle. Through the graph, we analyzed how the latitude and longitude values change over time in the collected GPS spoofing scenario data. There were two GPS spoofing scenarios: performing a spoofing attack from a specific point in time to the end and performing a spoofing attack intermittently. As a result of the analysis, we found that the third vehicle node perceived that the location information of the target vehicle changed continuously as shown in Figure 11.

As a result of analyzing the GPS values of the V2X messages that the simulation node collected, we found changes in the data when the GPS spoofing was running, as shown in Figure 12. Various V2X message components, such as heading, speed, and message count, can be used besides GPS data.

Although tests were performed by creating simple GPS spoofing scenarios, more diverse and sophisticated attack scenarios can be tested using the simulation environment proposed in this study. Since the virtual machine-linked simulation environment facilitates data transmission and collection, it can be used in various ways. For example, the performance of the misbehavior detection system can be verified by executing the attack scenarios after applying the security system to the virtual machine, or the weaknesses can be supplemented by analyzing the collected attack data. Furthermore, the weaknesses and the routes of cyberattacks performed up to the V2X network can be tested comprehensively using not only V2X messages but also the vulnerabilities of the linked virtual machine’s OS.

## 5. Security Evaluation Using Virtual Machine-Linked Simulation Environment

With the progress made in automobiles, various systems have been mounted on vehicles, increasing contact points with the outside world. This means that attackers have more ways of accessing vehicles. With the growing possibility of cyberattacks, vehicles’ cybersecurity standards, such as WP.29 R155 and ISO/SAE 21434, have emerged. The key is to identify and test threats to vehicles to prevent cyberattacks in advance in order to make vehicles safer. The factors of security evaluation for the safety of vehicles include standard conformity and misbehavior detection capability. In this section, we describe four types of test cases based on OmniAir’s SAE J2945/1 [36] test specification and conduct tests for four security evaluation scenarios through the simulation environment using a virtual machine.

The security evaluation scenarios proposed in this paper refer to the existing standards and evaluation methods. In addition, it was implemented using the simulation environment in Section 4 which satisfies the implementation requirements that are presented in Section 3. The following Table 3 presents the descriptions of the four scenarios. The proposed security evaluation scenarios can be simulated only for WAVE communication by using CANoe Option.Car2X’s WAVE-based simulation. Therefore, for security evaluation based on C-V2X and ITS-G5, equipment and simulation tools compatible with the relevant standards must be used. Since the proposed scenarios can evaluate misbehaviors that can commonly occur in V2X communication, it is possible to modify and utilize some of the scenarios to solve the limitations.

### 5.1. Security Evaluation Scenarios

In this section, we create test sequences for four security evaluation scenarios using the test purpose (TP) defined in the EG 202 798 [37] standard and conduct the tests in the simulation environment. The TP consists of the identification number, test objective, and procedure, and the test configuration types include the standard conformity, certificate, and event situation. Table 4 describes each item of TP.

#### 5.1.1. Security Evaluation Scenario 1

Scenario 1 is for verifying the data conformity, certificate, and certificate digest of the components of BSM. Table 5 shows the test sequence of Scenario 1.

Scenario 1 is verifying the BSM’s validity and certificate digest value. The verdict is “pass” only if the data and digest of the BSM are valid. It is a “fail” if any one item is not satisfied. To execute Evaluation Scenario 1, a certificate that the evaluation target node could use was generated. Two types of certificates were generated: trusted and untrusted. Figure 13 shows the information of trusted/untrusted certificates created through CANoe.

When the evaluation target node sends BSMs using a trusted and untrusted certificate, the verification node receives the BSMs. It verifies their validity through the standard conformity of the BSM core data and the digest values of the certificate. If both items are valid, the verdict is “pass”; if not, the verdict is “fail” as shown in Figure 14.

Through this scenario, a driving vehicle can verify BSMs and certificates received from other vehicles. When Scenario 1 is executed, it is possible to perform SAE J2735 BSM standard conformance and IEEE 1609.2-based certificate validity verification for messages received from neighboring vehicles at once.

#### 5.1.2. Security Evaluation Scenario 2

Scenario 2 tests whether *DE_VehicleEventFlags* is included only when an event occurs. The occurrence time of the message that includes the event information and the occurrence time of the message that does not include the event information after removing the event are verified according to the standard. Table 6 shows the test sequence of Scenario 2. *vEventDetectLatency* is a parameter defined 250 ms in the SAE J2945/1 standard. This scenario includes a process of validating whether a BSM message including *DE_VehicleEventFlags* is received within *vEventDetectLatency* time when an event occurs.

The terminal node of evaluation generates a vehicle event while driving according to the route. Here, we calculate the time to generate the last BSM before the occurrence of the vehicle event and generate a BSM containing *DE_VehicleEventFlags* after the event occurrence. Then, when *DE_VehicleEventFlags* is removed, the generation time of the BSM containing the event information for the last time and the generation time of the BSM with *DE_VehicleEventFlags* removed is calculated. Figure 15 is a screen that implements Scenario 2 in CANoe and displays the verification results as “pass” and “fail”.

Through this scenario, it is possible to evaluate the conformity of event generation in SAE J2735 and SAE J2945/1 standards. In particular, when an event, such as an accident, occurs, it is possible to verify whether or not DuT can propagate radio waves to the surroundings within a predetermined time.

#### 5.1.3. Security Evaluation Scenario 3

Scenario 3 is an evaluation scenario used to verify whether IUT has the misbehavior detection capability. There are various attack methods, but in this study, we assumed a situation where a malicious BSM is sent in the attack scenario. We created a simple attack scenario in which arbitrary speed values are sent using the same address as a normal driving vehicle. Table 7 shows the test sequence of Scenario 3.

Under normal driving circumstances, the vehicle’s speed changes smoothly. Therefore, a sudden change in speed within a certain time can be regarded as an attack. Based on this, we created a simple IDS that recognizes the attack when messages regarded as abnormal data occur more than a certain number of times within a certain time. It was applied to the evaluation target node. Then, the verification node generated a BSM with a random speed value and sent it to the evaluation target node to verify whether the evaluation target node would detect the misbehavior. Figure 16 shows a result that generates a warning when random speed data occurs in Scenario 3, which is regarded to be an attack.

A simple attack was performed in Evaluation Scenario 3, but there are more diverse and specific attacks in the real V2X environment. Therefore, sufficient testing and verification through various types of attacks are required to verify the misbehavior detection ability of the IDS of the actual evaluation target node.

#### 5.1.4. Security Evaluation Scenario 4

Scenario 4 evaluates the standard conformity by having the evaluation target send a message containing the whole certificate rather than the certificate digest value when *DE_VehicleEventFlags* occurs. The certificate type of the BSM sent by the evaluation target node after the occurrence of *DE_VehicleEventFlags* is verified. Table 8 shows the test sequence of Scenario 4.

*DE_VehicleEventFlags* was activated by triggering an event in the normal driving vehicle to test Evaluation Scenario 4. Here, the signer type of the BSM sent by the evaluation target node is verified at the verification node. Figure 17 shows the result of verifying the certificate signer type of the BSM message received for evaluation in CANoe, according to Scenario 4. When an event occurs, it is a “fail” if the signature included in the BSM is the “digest” value, and a “pass” if the entire “certificate” is included.

This scenario can be linked to Scenario 2. Scenario 2 verifies whether the message generates *DE_VehicleEventFlags*, and Scenario 4 verifies the certificate signer type of the message, including *DE_VehicleEventFlags*. If the BSM normally generated by event, the signer type of the message will be “certificate”, not the “digest”. Through this, the validity of the certificate of the event propagation BSM can be verified.

## 6. Conclusions and Future Research

With the advancement of automotive technology, various systems have been mounted on vehicles for user convenience. At the same time, the number of external contact points at which the vehicle can be accessed has increased. This means that the number of attack points at which attackers can attack vehicles has increased. The importance of vehicle cybersecurity has increased, since vehicle attacks can directly impact drivers, surrounding vehicles, and people. UNECE WP.29 and ISO/SAE 21434 recommend the application of cybersecurity elements to vehicles. After applying security elements, various security tests are needed to verify whether the system is fully operating and whether there are any errors. However, there are time, cost, and safety issues when performing tests in a real-world environment. Hence, simulations are used to perform various tests in advance, but there are difficulties in performing some tests because they are far from the real-world environment. Furthermore, we need a method of evaluating whether the security technology has been applied at the application level, rather than conformity testing, for processing misbehavior.

In this study, we described a test platform that linked a virtual machine and the simulation equipment to overcome the limitations of conventional simulations. We also listed the methods of using the actual test platform using four security evaluation scenarios. Using the test platform proposed in this paper, one can secure time, cost efficiency, and safety. Furthermore, results can be obtained with various tests, since security threats and misbehaviors in the real-world environment can be tested. Data collection, analysis, and misbehavior detection of IDS can be performed without being constrained by programming languages, since the data generated by the simulation equipment can be processed from the outside.

In the future, we plan to research to develop additional scenarios besides the four security evaluation scenarios used in this paper. Here, each scenario will be created more specifically to create practical scenarios for security evaluation. Currently, the research focuses on V2V, but we plan to develop test cases for various security threats that may occur in the Security Credential Management System (SCMS) and vehicle driving scenarios, as well as messages used for V2X, such as SPaT and MAP messages.

## Figures and Tables

**Figure 1 sensors-23-01421-f001:**
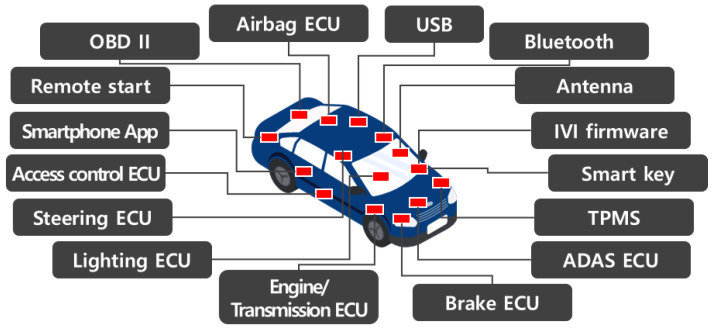
Connected car internal/external interfaces.

**Figure 2 sensors-23-01421-f002:**
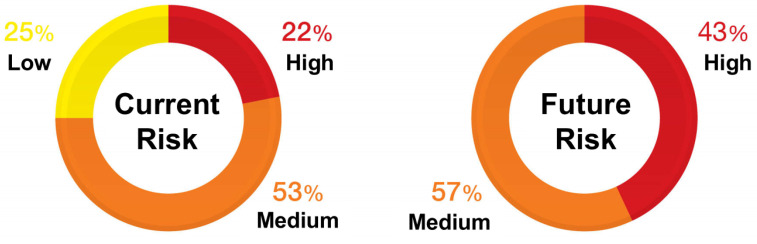
Current and future risks to the attack vector of connected cars. In the future, most attack vectors will shift to medium–high risk threats.

**Figure 3 sensors-23-01421-f003:**
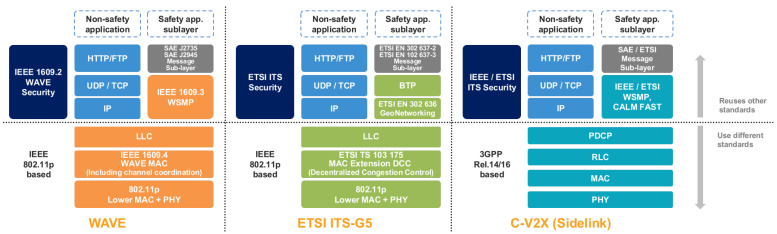
Comparison of ITS stacks by V2X protocol. WAVE uses U.S. standards, and ETSI ITS-G5 uses European standards. In C-V2X, the lower-level layer uses 3GPP specifications, and the higher-level layer can apply U.S. and European standards.

**Figure 4 sensors-23-01421-f004:**
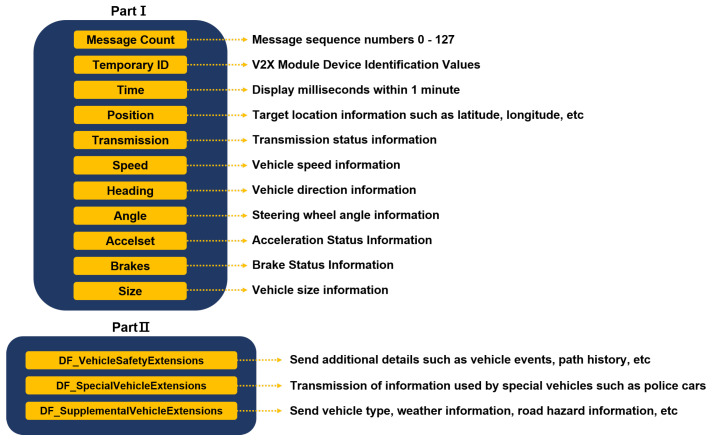
Structure of the BSM message Part I; Part II defined by SAE J2735.

**Figure 5 sensors-23-01421-f005:**
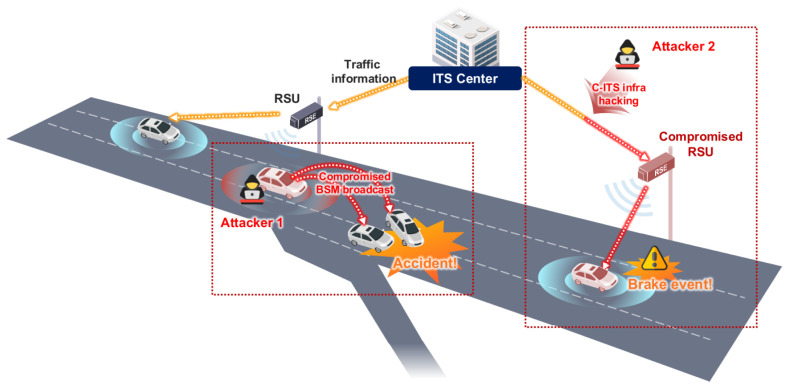
Examples of an attacks using V2X. Attacker 1 is the attacker’s vehicle and causes an accident by sending a compromised BSM to near vehicles. Attacker 2 hacks the C-ITS infrastructure or RSU to send false information to the vehicle or force events, such as hard braking.

**Figure 6 sensors-23-01421-f006:**
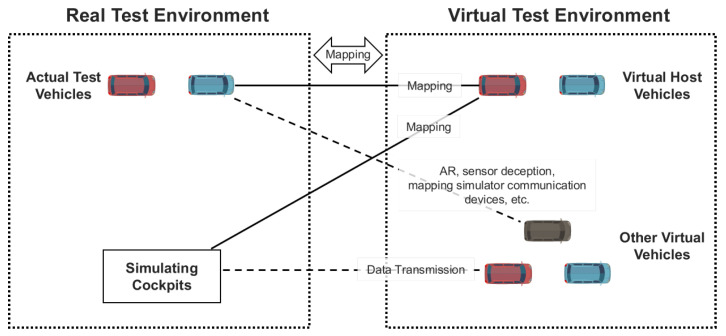
Environment configuration for parallel testing proposed by J. Wang et al. Application level security evaluation should be performed in an environment where a real device is mapped to a virtual test environment, as shown the actual test vehicle in this figure.

**Figure 7 sensors-23-01421-f007:**
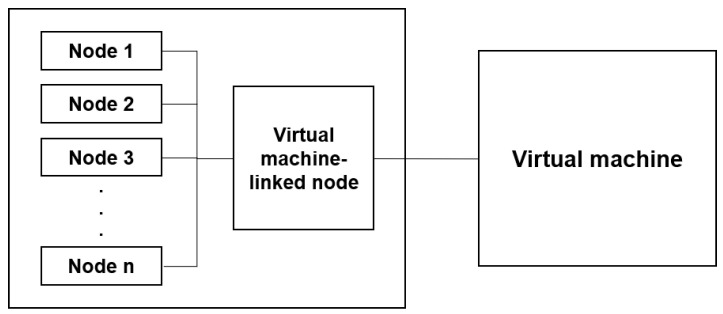
This figure shows the basic architecture of the virtual machine-based simulation node interoperation test environment configuration proposed in this paper. The user can configure the simulation nodes and the virtual machine as n:n according to the test situation.

**Figure 8 sensors-23-01421-f008:**
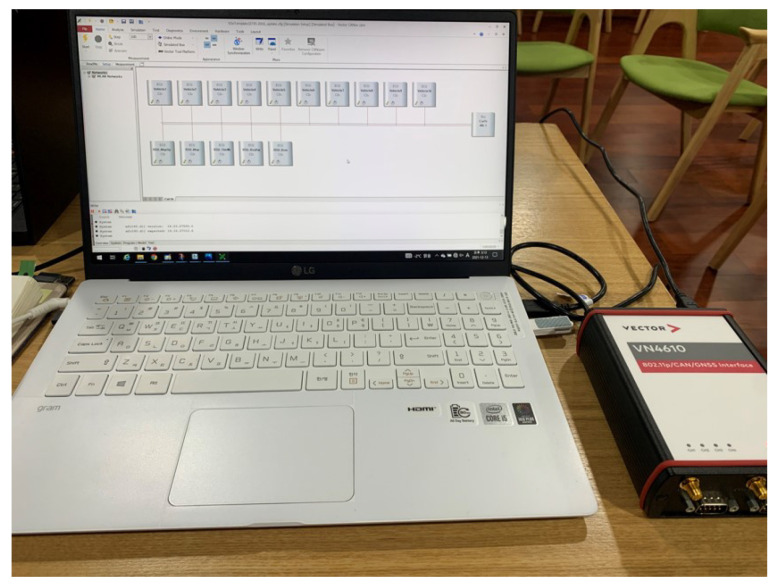
Environment composed of laboratory laptop and CANoe.

**Figure 9 sensors-23-01421-f009:**
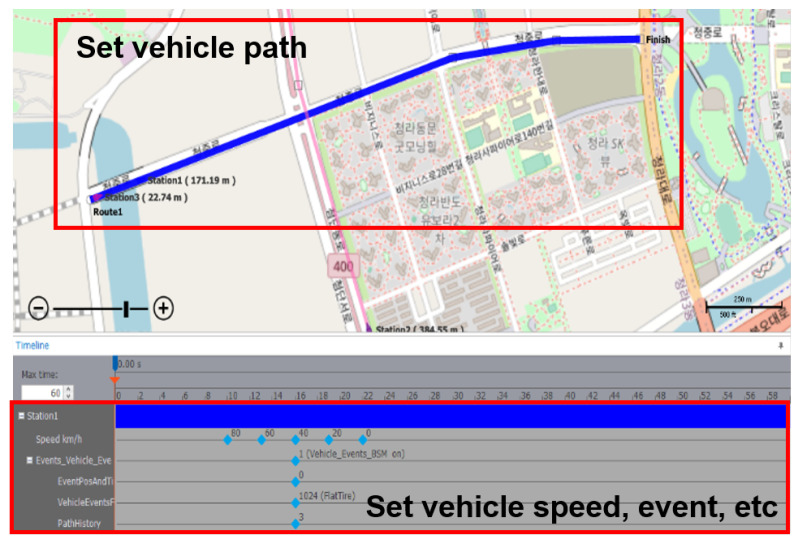
The simulation editor of the simulation equipment used in this paper. Using the scenario editor, the user can specify vehicle nodes’ routes, speeds, and events. When the simulation is executed, each vehicle node drives according to the scenario specified by the user.

**Figure 10 sensors-23-01421-f010:**
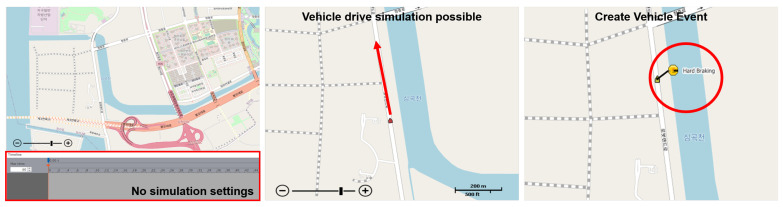
This figure shows driving by sending V2X data to a simulation node linked to the virtual machine and generating a vehicle event (hard braking) without using the scenario editor. The simulation node generates a message according to the standard specifications for the data received from the virtual machine and sends it to the simulation network. The user can create scenarios for specific situations or events according to the test cases.

**Figure 11 sensors-23-01421-f011:**
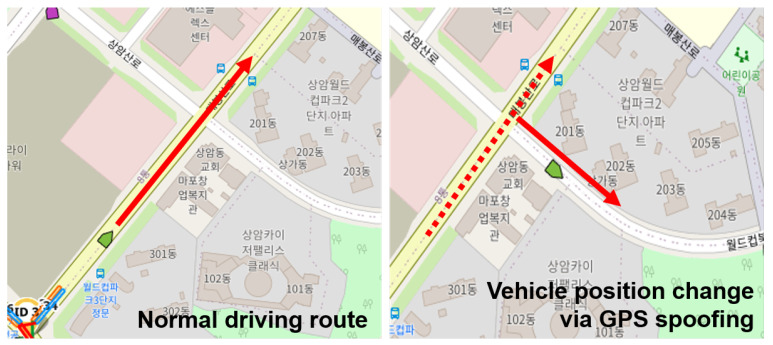
This figure shows a GPS spoofing scenario in which the GPS values are changed using the same address and ID in a vehicle node driving usually. The third vehicle perceives that the GPS information of the vehicle, with the pertinent ID, changes continuously, as shown in the figure.

**Figure 12 sensors-23-01421-f012:**
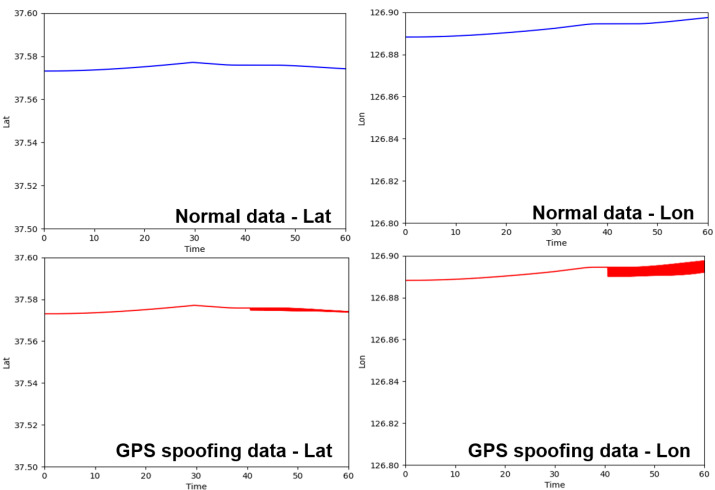
The graphs are the results of collecting and analyzing V2X messages generated in the simulation network after performing a GPS spoofing attack. With the occurrence time of V2X messages on the x-axis and the lat and lon values on the y-axis, the graphs show the changes when the GPS spoofing attack occurred. When driving normally, the values changed smoothly according to the driving route, but when the GPS spoofing occurred, the GPS value moved rapidly.

**Figure 13 sensors-23-01421-f013:**
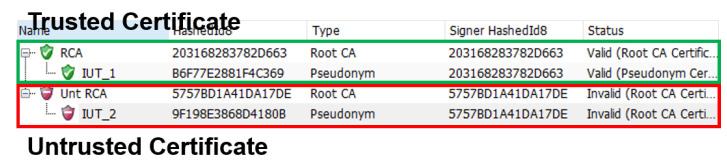
Certificates that can be used when the evaluation target node sends a BSM. The evaluation target node sends a BSM using the specified certificate. Two types of certificates (a trusted and untrusted certificate) were generated to show the pass and fail scenarios.

**Figure 14 sensors-23-01421-f014:**
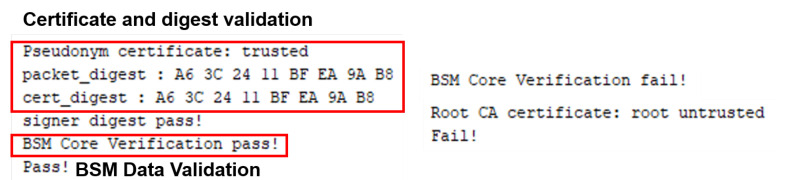
Result of verifying the core data and certificate of the BSM that the evaluation target node sent. The verification is passed only if both items are valid.

**Figure 15 sensors-23-01421-f015:**
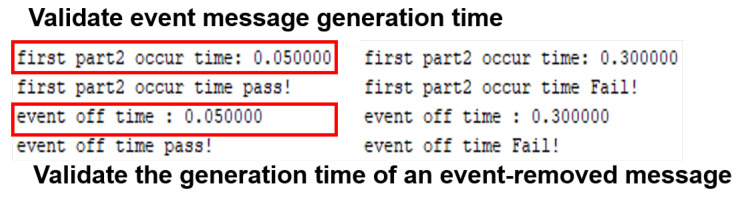
Pass/fail is determined based on the generation time of BSM before and after the occurrence of the event of the terminal node of the evaluation. In this figure, the time taken from generation to reception of part II data is measured and displayed, and result is "pass" if transmitted within 50 ms. And, if the occurrence time of part II event ends is within 250 ms (vEventDetectLatency), the result is "pass".

**Figure 16 sensors-23-01421-f016:**
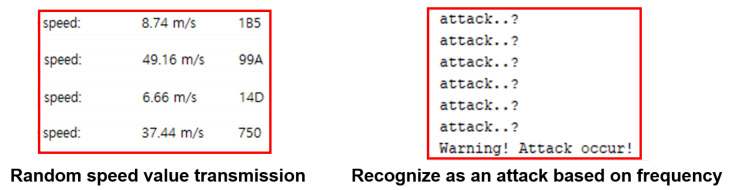
In this figure, the evaluation target node that received the BSMs sent by the verification node recognizes that the changes in the speed value are not normal and recognizes the attack, since the changes occurred more than a certain number of times within a certain time. In this study, we used simple misbehavior and IDS, but in real tests, the misbehavior detection ability of the IDS should be verified using various types of attacks.

**Figure 17 sensors-23-01421-f017:**
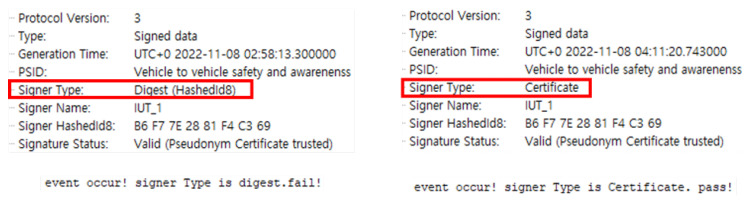
The verification node checks the signer type of the BSM sent by the evaluation target node. When an event occurs, the signer type must be “certificate”. If it is “digest”, the verdict is “fail”. The figure on the left shows an event occurred but the signer type is “digest”, and the result is “fail”. The figure on the right shows that the signer type is “certificate” when an event occurs, and the result is “pass”.

**Table 1 sensors-23-01421-t001:** The main message set of SAE J2735.

Message		Description
**BSM**	Basic Safety Message	Vehicle V2X safety information
**SPaT**	Signal Phase and Timing	Traffic signal and timing information
**MAP**	Map Data	Intersection and road lane information
**TIM**	Traveler Information Message	For sending advisory and road sign information
**RSA**	Road Side Alert	For the alerting of nearby hazards information
**PVD**	Probe Vehicle Data	For collecting a vehicle’s traveling information
**TAM**	Toll Advertisement Message	V2X-based fee collection data with SAE J3217 [20]
**SDSM**	Sensor Data Sharing Message	Reserved for future use

**Table 2 sensors-23-01421-t002:** Comparison of requirements for V2X simulation and testing tools for security evaluation scenario development.

	Lokaj et al. [28]	Veins (with VeReMi) [32]	Spirent V2X Test [33]	Keysight Call Simulator [34]	Vector CANoe [35]
**Target**	V2X simulation research		Commercial evaluation tool		
**Purpose**	Validation of message interoperability	Simulation traffic and message validation	Validation test	Device test	Device test
**ITS stack**	Δ (Only V2X message)	O (Provided as a module)	O (Based on Call simulator)	O (Provide full stack)	O (V2X messages require additional implementation)
**Security** **evaluation**	Δ (Only message conformance)	Δ (Only for specified data)	Δ (Only standard conformance)	N	Δ (Feature implementation required)
**Acceptance of** **external** **message**	O	Δ (Use VeReMi set)	O (Can receive external device message)	O (Can receive external device message)	O (Can receive external device message)
**Application** **message** **handling**	N (Only check message interoperability)	Δ (Only ITS application)	Δ (Only handle message for test)	N	O (Can implement send/receive messages)
**Creation** **abnormal** **scenario**	N	N	N	N	Δ (Provide implementation environment only)

O: Offer, Δ: Partially offer, N: Not offer.

**Table 3 sensors-23-01421-t003:** Description of proposed scenarios.

No.	Description	Reference	
**Scenario 1**	BSM conformance and certificate validation test	SAE J2735 [19]	6.10 DF_BSMcoreData
		IEEE 1609.2 [10]	6.3 Secured protocol data units
**Scenario 2**	Vehicle event flag validation test	SAE J2735 [19]	7.221 DE_VehicleEventFlags
		SAE J2945/1 [36]	Table 17 - SAE J2735 requirements
			6.3.1 BSM Contents (BSMCONT)
			6.3.6.15 DE_VehicleEventFlags
**Scenario 3**	Abnormal speed detection test	UN Regulation 155 [1]	7.2.2.4 Specifications
		ISO/SAE 21434 [3]	10.4.2 Integration verification [RC-10-12]
			11 Cybersecurity validation [RQ-11-01]
**Scenario 4**	Certificate type validation test in event situation	SAE J2945/1 [36]	6.5.2 BSM Signing (BSMSIGN)

**Table 4 sensors-23-01421-t004:** Description of each item in TP.

Item	Description
TP ID	The unique identifier of TP
Test object	A simple description of the test objective and goal
References	Reference standards for conformity requirements
Test configuration	Test configuration type of TP
Pre-test conditions	The initial conditions that the IUT must apply to apply TP
Test sequence	Test procedure
Stimulus	Refers to an event generated to allow the IUT to perform a specific task
Check	Determining whether the conditions are appropriate
Configuration	Refers to the IUT operation in the test stage
Verify	Verifying whether the IUT operates as expected. Classified as pass/fail
Procedure	A specific action is instructed, e.g., Repeat steps 1 to 4

**Table 5 sensors-23-01421-t005:** Test sequence of Evaluation Scenario 1.

Test Sequence			
**Step**	**Type**	**Description**	**Verdict**
1	Stimulus	A BSM is sent	
2	Verify	It is checked whether the BSM contains the certificate digest	pass/fail
3	Verify	Verification of the certificate validity	pass/fail
4	Verify	Verifying whether BSM core data are included	pass/fail
5	Verify	Verifying the validity of BSM core data values	pass/fail
6	Configure	Renewing the certificate	
7	Procedure	Repeating steps 1–4	

**Table 6 sensors-23-01421-t006:** Test sequence of Evaluation Scenario 2.

Test Sequence			
**Step**	**Type**	**Description**	**Verdict**
1	Verify	Verifying whether the BSM is sent with *DE_VehicleEventFlags*	pass/fail
2	Stimulus	Occurrence of one of the vehicle events	pass/fail
3	Verify	Verifying whether the BSM containing *DE_VehicleEventFlags* is sent within 50 ms	pass/fail
4	Verify	Verifying whether the data corresponding to the event is sent in the BSM containing *DE_VehicleEventFlags*	pass/fail
5	Stimulus	Removal of *DE_VehicleEventFlags*	
6	Verify	Verifying whether *DE_VehicleEventFlags* is included within *vEventDetectLatency*	pass/fail
7	Procedure	Repeating steps 2–6 for various events	

**Table 7 sensors-23-01421-t007:** Test sequence of Evaluation Scenario 3.

Test Sequence			
**Step**	**Type**	**Description**	**Verdict**
1	Configure	BSM is transmitted normally	
2	Stimulus	A BSM with a random speed value is sent, and the target receives the message	
3	Verify	Verifying whether the IUT receives the malicious message	pass/fail
4	Verify	Verifying whether the received malicious message is determined as misbehavior	pass/fail
5	Verify	Verifying whether the misbehavior information is reported	pass/fail

**Table 8 sensors-23-01421-t008:** Test sequence of Evaluation Scenario 4.

Test Sequence			
**Step**	**Type**	**Description**	**Verdict**
1	Configure	BSM is transmitted normally	
2	Stimulus	Occurrence of a vehicle event	
3	Verify	Verifying whether the data corresponding to the event is sent in the BSM containing *DE_VehicleEventFlags*	pass/fail
4	Verify	Verifying whether the certificate type of the message sent by the target is “certificate”	pass/fail

## Data Availability

Not applicable.

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
