# Peer review of "Simulation-Based Cybersecurity Testing and Evaluation Method for Connected Car V2X Application Using Virtual Machine"

_sensors, 2023, doi:10.3390/s23031421_

Round 1
Reviewer 1 Report
1. the fist line of abstract "In connected cars, security vulnerabilities that did not exist before can occur due to the connectivity between the vehicle’s internal network and the outside" - make asimple sentence rather than conjugate sentence.
2. do put the comparitive results in a tabular form.
3. 5G and IoT are the dominant force in V2V communication. so it is advisable to include this in the introductiona nd banckground atleast.
4. A paragraph should be wriiten on as a concluding remarks obn all the observation is the section 5, sinc eteh conclusion is not reflecting it.
5. how could your study is unique from others reseracher contribution? there is no mentioning of this.
6. how vulnerable is teh current TCP/IP model on the threat a sdiscussed by you? there is no mentioning of this aspects.
Author Response
Dear Reviewer,
Please see the attachment.
Sincerely,
Dae-Hwi Lee.

Reviewer 2 Report
In my view, this is a really intriguing work, and I have no doubt that its readers will find it interesting as well. Some minor problems that I see in the article are as follows:
1- Please provide more explanations than the introduction and statement of the problem in the beginning and also discuss and consider more references, and update references in the Introduction. Some references that may be of interest to your readers are suggested as follows:
10.1109/DASC/PiCom/CBDCom/Cy55231.2022.9927793
10.1109/CoDIT55151.2022.9804014
10.1109/TR.2022.3159664
https://doi.org/10.1016/j.vehcom.2021.100428
2- In section 3: Why you consider 3 requirements of implementation (3.2.1,3.2.2,3.2.3) in your study?
3 In section 4.1.: Authors mentioned “Since each node implemented in a typical commercial V2X communication simulation exists only within the simulation, it is not connected to the external ITS infrastructure. Please consider a reference for this statement.
network, unlike the ITS components in the real-world environment.
4- Page 11: Even though Section 5 is well-explained, authors need to consider more explanation about your scenarios.
5- It's suggested that the model's restrictions be included at the end of Section 5.
6- Minor English grammar and spelling checking is required.
Author Response

(The authors gave the same response as above.)

Round 2
Reviewer 1 Report
all the comments have been successfully justified.